# Burden and resources in caregivers of people with multiple sclerosis: A qualitative study

Stefano Benini[1], Erika Pellegrini[1], Carlo Descovich[1], Alessandra Lugaresi[2,3]*

**1** AUSL di Bologna, Bologna, Italia, **2** IRCCS Istituto delle Scienze Neurologiche di Bologna, Bologna, Italia, **3** Dipartimento di Scienze Biomediche e Neuromotorie, Università di Bologna, Bologna, Italia

* alessandra.lugaresi2@unibo.it

**Data Availability Statement:** Data are available from Zenodo: https://zenodo.org/record/7788479#. ZCa-SWBBzIU.

**Funding:** The publication of this article was financially supported by the "Ricerca Corrente" funding from the Italian Ministry of Health. No

## Abstract

### Background

Caregivers of people with Multiple Sclerosis are required to provide ongoing assistance especially during the advanced stages of the disease. They have to manage interventions and assume responsibilities which significantly impact both their personal quality of life and family's dynamics.

### Objective

A qualitative phenomenological study was carried out to understand the experience of burden in caregivers and their resources to manage it. The study also explores how healthcare services involved in the Multiple Sclerosis Clinical Pathway respond to the needs of well-being of patients and family members.

### Methods

17 caregivers were involved in focus groups and in semi-structured individual interviews.

### Results

Fatigue is experienced by all respondents and it starts when physical disabilities increase or when people become aware of them. Many caregivers declare that they refer to intrinsic (love towards their relatives, patience and dedication) or extrinsic (family members, hobbies) resources to cope with the burden of assistance. Patient associations and the Multiple Sclerosis Clinical Pathway play a significant role in supporting caregivers.

### Conclusions

Fatigue, loneliness, and isolation are experienced by caregivers and strongly affect their quality of life and health status. The study highlights caregivers' need to reconcile working times with care times, to give more space to self-care and to have moments to share their experiences with someone else. These needs should be at the core of health policies in order to avoid physical and emotional breakdowns which could lead to the rupture of the relational balance on which home care is based.

additional external funding was received for this study.

**Competing interests:** SB, EP and CD have declared that no competing interests exist. AL has the following competing interests: has served as a Biogen, Bristol Myers Squibb, Merck Serono, Novartis, Roche, Sanofi/ Genzyme and Teva Advisory Board Member. She received congress and travel/accommodation expense compensations or speaker honoraria from Alexion, Biogen, Merck Serono, Mylan, Novartis, Roche, Sanofi/Genzyme, Teva and Fondazione Italiana Sclerosi Multipla (FISM). Her institutions received research grants from Novartis and Sanofi/ Genzyme.

## 1. Introduction

Multiple sclerosis (MS) is often characterized by increasing disability over time, despite treatment. The variable disease course and symptoms require a multidisciplinary approach and MS Care Units include different healthcare professionals, who share standardized procedures. The MS Care Unit should offer a personalized pathway, offering specific case and care managers, who can provide information, support and advice to the individual MS patient [1].

Highly specialised interventions prevail in the initial stages of the disease. This initial care is provided mainly in the hospital setting. As the disease progresses, treatment strategies are integrated with social care and rehabilitation interventions, mainly at home or in its proximity.

The opportunity for the person with multiple sclerosis (PwMS) to stay at home is made possible thanks to the implementation of a strong synergistic network of cures and treatments involving health services, social care workers, volunteers from associations and family members.

Family members are often required to provide ongoing assistance, especially in the advanced stages of the disease.

Informal caregiving is provided by spouses in up to 70% of PwMS, with the remaining of care usually provided by other family members and friends [2].

Caregivers have to manage multiple interventions and assume responsibilities that significantly impact both their personal quality of life, and the family's dynamics [3].

Caring for someone with MS can be emotionally and physically demanding and can lead to considerable stress among carers, negatively affecting their health and well-being [4]. Indeed, there is consistent evidence in the quantitative literature that MS carers' quality of life (QoL) is poor [5]. Previous studies have demonstrated that high levels of carer burden and anxiety are strongly associated with reduced QoL [6,7]. Severity of MS symptoms (e.g., impaired mobility, bladder problems, cognitive impairment, and depression), increased carer demands. Longer hours of caregiving, are also strong predictors of poor QoL and psychological ill-being in MS carers [8–11].

Compared to caregivers of individuals with other chronic conditions, caregivers of PwMS often start caregiving earlier in life and may remain in this role for decades, often with increasing projected disability [12].

For example, over 80% of family caregivers of individuals with MS have reported that time required to care for their relatives meaningfully interferes with their lives including social role functioning (i.e., obligations with family, friends, and career) [13].

In the literature, unpaid caregivers are defined as *hidden patients* because of the physical, emotional and economic burdens they have to bear to assist their loved one, with or without the contribution of social services [14]. Addressing and accepting all the issues related to the presence of an unpredictable and long-lasting progressive disease is a source of great stress for the family members [15]. This stress increases as the physical and cognitive disabilities of the loved one progress [16].

In fact, continuous adaptation to the increasing needs for time and care leads family members to change their lifestyle and habits [17]. This process generates high levels of fatigue which, in turn, is associated with mood disorders. These can be expressed by experiencing: mixed feelings, nervous strain and depression with episodes of anger and hostility expressed towards the family member with MS [18].

The caregiver, more than any other figure who enters the field in social health [19] plays a key role in providing moral assistance and comfort to the PwMS. There must be strong investment in supporting the caregiver providing the multifaceted burden of care.

In this regard, the Clinical Pathway (CPW), for PwMS in the Local Health Authority (AUSL) of Bologna, was developed with the intention to provide concrete answers to the care and welfare needs not only of the persons affected, but also of their relevant others.

The aim of the CPW is to translate the best research evidence gathered into clinical practice recommendations adapted to local resources and specific clinical processes in order establish standard procedures for diagnosis, treatment and care for PwMS and to coordinate all parties involved in the treatment of the disease and working in large territorial areas such as Bologna.

The CPW is subdivided into three phases—1. Access and diagnosis, 2. Taking charge / healthcare in the phase of mild to moderate disability and 3. Taking charge / healthcare in the phase of severe disability. The CPW offers multidimensional care for the patient and relevant others through validated, integrated and multi-professional care plans, and pays particular attention to the continuity of social-health care and home care.

The first studies on the burden of care on the caregiver in MS date back to slightly over 20 years ago, and most of the data has been collected to understand the correlation between caregiver status and the well-being of the patient. Nevertheless it is necessary to conduct further studies to better understand the components of the burden of care and to study the effectiveness of interventions aimed at supporting caregivers in their task.

The main objectives of this research are to: a) understand the burden of care on caregivers and their the ensuing experiences, b) assess which resources the carers rely on to manage the burden of care.

How the healthcare services and workers involved in the MS CPW respond to the welfare needs of patients and family members will be the object of future research.

## 2. Methods

Among different approaches in this study we used a hermeneutic phenomenological methodology, [20,21] because, this approach focuses on people's everyday life experiences and subjective phenomena and seeks to understand the meaning of these experiences [22]. The aim of this study is to identify the most relevant issues, in terms of burden and resources, needed by PwMS caregivers. Interviews included both "what" they had experienced and "how" they had coped with it [23].

This method has been successfully used in previous studies conducted with caregivers and appears as the one best suited to a qualitative approach [24–27].

### 2.1 Participants and sampling

The caregivers of people who are in the care of the MS CPW, and who attend the local Bologna section of Italian Multiple Sclerosis Association (AISM), took part in the survey of the AUSL of Bologna. We chose AISM because in Bologna it is the association with the highest number of members and caregivers. The recruitment of eligible caregivers took place, for convenience, at the local AISM section, taking into account their willingness to participate in the survey. Caregivers were informed by the volunteers attending the AISM section in Bologna. Interviews were held until no new themes emerged (data saturation). Inclusion criteria were that caregivers were: 1) the primary informal caregiver of a PwMS; 2) 18 years of age or over; 3) Italian speaking; 4) willing to participate in the study and to sign the informed consent form and 5) had cared for a PwMS for at least 5 yr. Carers of people with a follow-up from diagnosis shorter than 5 yr were excluded.

### 2.2 Data collection

After caregiver enrolment, the investigator asked them if they would prefer to be interviewed within a focus group, held at the AISM headquarters, or individually, at home and appointments were made accordingly. We used the triangulation with two methods of data collection —Focus group (FG) and In-Depth interviews (IDIs)—to develop a comprehensive

understanding of caregivers experiences [28]. We decided to use both FG and IDIs interviews for two reasons. The first was to enhance participation of a broader spectrum of eligible caregivers who might not otherwise be able to participate if a single method of data collection were selected. The second reason was to increase the validity of study findings through collection and triangulation of data [29].

After they had signed the informed consent form, the investigator invited the caregiver to give a narrative description of their feelings and experiences by asking the following open ended questions: "You take care of a PwMS; could you please tell me about your experience of giving care to him/her? You can answer through an image, a colour or an adjective". "What does it mean for you to take care of a PwMS?" "What's your experience in the MS CPW of the AUSL of Bologna and in AISM?" The same questions were asked during the focus group and interviews.

During the interview or in the focus group, the investigator adopted an empathetic attitude towards the caregivers, encouraging them to talk. When caregivers seemed to have nothing more to say, the researcher asked if they would like to add anything further. This process continued until the caregiver had nothing more to say. Each interview took between 20–60 minutes, and all interviews were audio-recorded. At the end of the interview, the investigator collected the caregiver's socio-demographic information, such as age, gender and data from patients such as age, gender and patient's illness duration.

## 2.3. Data analysis

All interviews were transcribed verbatim. Interviews were analysed using Colaizzi's descriptive analysis framework [30], divided into six analytical steps. (a) Reading of interviews by two researchers (SB, EP) who listened to the audio and re-read transcripts several times to become immersed in the data; (b) Identification of significant statements; (c) Formulation and validation of meanings through team discussions (SB, EP, CD); (d) Organisation of each significant statement into meaning units and sub-themes into major themes; (e) Check of meaning units, sub-themes and themes by an expert qualitative researcher (SB); (f) Definition of overarching statements to summarise the participant's lived experience.

The researchers kept a research diary with methodological and reflective notes. They documented all of the passages and critical junctions of the work, trying to be conscious of their own pre-understandings and adopted an approach to the data avoiding preconceived perspectives.

The research team used several strategies to ensure the accuracy of the findings. The analytic team met regularly throughout the iterative process of data analysis to reflect on ways individual values, experiences, or preconceptions might influence or create bias in interpretations of the data may have informed perceptions of participant statements and emerging themes. In addition, the analytic team wrote memos throughout the data analysis, noting observations and reactions to the interview responses. Finally, the last author (AL), who had not previously participated in data analysis, conducted an independent review of the findings and coding documents to assess the trustworthiness of the analysis.

## 2.4. Ethical considerations

The study has been conducted in full compliance with relevant data protection regulations.

The protocol was submitted to the attention of the local ethics committee (Comitato Etico di Area Vasta Emilia Centro of the Regione Emilia-Romagna (CE-AVEC)) which replied that no authorization was required as no personal data was processed in this study.

Each family member has been informed about the purpose of the study and signed the consent form for participation in the research prepared by the AUSL of Bologna for qualitative studies.

## 3. Results

The study started in July 2019 and ended in February 2021.

Of the 21 caregivers initially willing to participate in the study, 4 were no longer available due to deterioration of the health conditions of their relatives; of the remaining 17, 9 decided to participate in the focus group and 8 to the interviews. We held 1 single focus group due to the manageable number of participants.

Of the 17 family members involved, 14 (82.35%) were females and 3 males, with an average age of 65 years (range 49–81). There were 8 wives, 1 husband, 2 daughters, 4 siblings (3 sisters and 1 brother) and 2 parents (1 father and 1 mother) of people who had received a diagnosis of MS from 5 to 39 years earlier (Table 1). Most of the patients have a mild-to-moderate disability (EDSS 3.5–6.0), 3 have a moderate-to-severe disability(EDSS>6.0) and are in charge of the MS CPW of the Bologna Local Health Authority. Almost all of the participants live in the same home as the family member with MS.

From the analysis of the interviews, it was possible to identify the following themes: a) The burden of assistance and the effect of the effort; b) States of mind and concerns; c) Resources to cope with the assistance burden; d) Take charge by the MS CPW of the AUSL City of Bologna; e) The role of the Patient Association; f) The needs of family members.

The analysis of the 6 themes is reported in the following paragraphs.

### Theme a: The burden of assistance and the effect of the effort

Feedback was solicited through the use of metaphorical language using the expression of an image, colour or adjective. Family members represented the burden of care, on the one hand,

**Table 1. Characteristics of people interviewed.**

| Interviewed Participant (P) Code | Age | Degree of relationship | Age of relative with MS | Age of relative with MS at diagnosis |
|---|---|---|---|---|
| P01 | 61 | Sister | 58 | 53 |
| P02 | 65 | Wife | 68 | 37 |
| P03 | 79 | Mother | 53 | 28 |
| P04 | 81 | Father | | |
| P05 | 67 | Husband | 61 | 35 |
| P06 | 70 | Wife | 66 | 31 |
| P07 | 53 | Wife | 54 | 26 |
| P08 | 67 | Wife | 66 | 27 |
| P09 | 73 | Sister | 62 | 26 |
| P10 | 74 | Wife | 77 | 48 |
| P11 | 63 | Brother | 52 | 26 |
| P12 | 59 | Daughter | 85 | 60 |
| P13 | 64 | Wife | 69 | 64 |
| P14 | 54 | Wife | 57 | 52 |
| P15 | 70 | Wife | 72 | 61 |
| P16 | 49 | Daughter | 78 | 56 |
| P17 | 58 | Sister | 52 | 33 |

as very heavy and challenging, as shown by the expressions: *"for me the burden of care is like an endless labyrinth"*(P9), *"The burden of care? Interesting. . ..I think the burden is. . . grey as tiring, Yes the burden is grey!"* (P17) or *"The colour of my burden is purple. . .Purple because the situation is disastrous, it is very heavy, our family is destroyed. We have a son in Sicily that we see so little (very few). . .* (P2). And on the other hand, as manageable, although related to the stages of the disease, expressed like *"I say blue when the situation is under control, but also grey and black when there are peak periods"* (P11). Although caring for a loved one at home is recognized as tiring, some gave a positive connotation to this burden *"for me it is not hard"* (P7) or *"I say yellow because of the positivity of the colour and because he is at home and I can care for him"* (P13).

Going into the details of the experiences gathered, the efforts expressed were amenable to 6 distinct issues, as depicted in Fig 1. The first and main issue was physical fatigue for the material aspects of assistance, mainly concerning personal hygiene (*washing*, *taking to the bathroom*, *changing diapers*, *dressing/undressing*), activities related to mobilization (*getting up/ putting to bed*, *transferring to the wheelchair*, *resettling in the wheelchair when slipping out*, *changing home due to the presence of architectural barriers*) and, sometimes, the administration of therapy. In many cases, physical fatigue was associated with the advancing age of the caregiver and the condition of the loved one. One family member expressed: *"I should be younger here, and he should be less heavy, because if he were slimmer it would require less effort. Now both in the morning, when he has to be lifted from the bed, and in the evening when he needs to be transferred from the wheelchair to bed, it is an effort difficult to manage"* (P15). The second issue was difficulty in accepting the disease, the change or disability of your loved one: *"I'm fatigued on a mental level, I am concerned because I remember the person in a way which is now different. . . he was dynamic and an active person. He was always ready to do anything. . ..and now he is bedridden or in a wheelchair. . .this is my burden. . ..my mind can't accept all this"* (P13). The third issue was fatigue resulting from the management of care and the sense of responsibility: *"My labor? I'm fatigued for all concerns: I have to make notes for everything: treatment, visits, that is, I have to take care and responsibilities for everything. Those are my main efforts!"* (P10). The fourth issue was fatigue due to concerns for the future: *". . . as long as I can, I will assist my child, because I'm his parent, but if I get sick, when I am old, and die, who will stay with him and take care of him?"* (P03). The fifth issue was fatigue due to having to

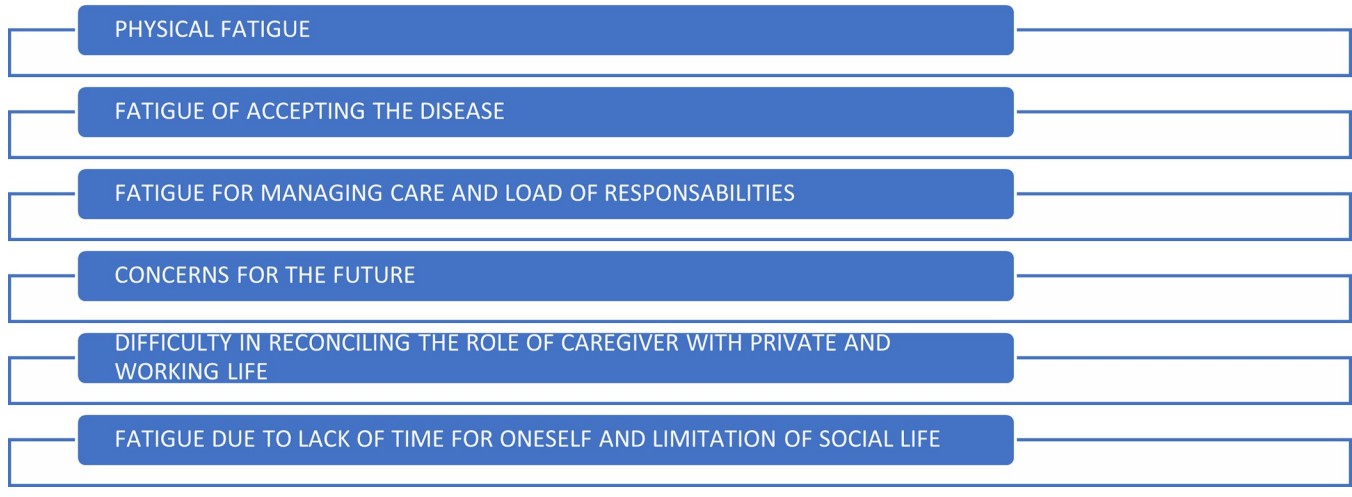

**Fig 1. The fatigue of caregivers in assistance.**

reconcile the role of caregiver with one's private life and work activity: "*Can I speak about my constant change of role? You are the caregiver, but later you go to work and you have to be efficient, productive. The one who reasons and who is attentive. This continuous ballet of emotions, this eternal commitment, is something that today, has become a lot harder to bear*" (P01). The sixth and final issue was fatigue due to lack of time for oneself and limitation of social life: "*Since the disease became so patent, I have not had a day off. I never have a truce, because I'm always there. Because she is alone, here I am. I can't stay more than 2–3 hours away from home, because she gets anxious*" (P05). Another family member says: "*I am smiling now because I like talking to a person I don't know, but in my life I don't have any social life anymore, also because when I get home I'm tired*" (P08).

Although the sense of fatigue is a constant experience in all interviewees, there are occasions when it takes on a particular weight. This is the case for the efforts related to the bureaucratic management of care and assistance, such as, when you have to organize an outing for a visit, or you have to fulfil bureaucratic tasks. In this regard, some family members underlined the difficulty in starting the process of getting the disability allowance, or managing administrative activities, such as paying contributions for a caregiver. Administrative burdens may create emotional costs especially when policy processes are experienced and stressful. A wife told us: "*It's so hard when I've to renew the disability certificate of my husband, . . ..few days before the assessment I start getting anxious, because I don't' know what will happen*" (P02).

The moments of fatigue are also felt in some phases of the day, such as during direct care activities such as getting up or going to bed, in mobilization and personal hygiene. The sense of fatigue takes on a particular weight even more so when the caregiver is in a moment of particular vulnerability: the reference was done, for example, to the frustration of not being able to live a normal daily life with their loved one, as told by a wife: "*When I go out and I see people of our age walking hand in hand, I get in a* rage" (P14); or in the conditions of illness / deficit of the caregiver himself: "*I'm afraid to get sick. . . Who will take care of my son when I'm not well? Who gets out of bed, for example, when I have back pain? In those cases I take anti-inflammatory drugs and I pray to recover as soon as possible*" (P03). Extraordinary events are particularly tiring, such as in the case of the worsening of the health of your loved one that involves a reorganization of daily activities. "*My worst fatigues are two: the first is to see my husband get worse with MS. . . . It is heartbreaking to see him lose his autonomy; the second is not being able to sleep in the same bed*" (P08).

Inquiring about the period of onset of the state of fatigue, some say they have experienced it since the onset of the disease, especially with reference to the emotions triggered by the communication of the diagnosis: "*When the neurologist communicated us that our son had MS I felt like dying. . .I remember I told "It's not possible"*" (P04). Others, on the other hand, started to feel fatigued along the course of the disease especially when physical disabilities increased, or when they became aware of the disabling course of the disease.

## Theme b: States of mind and concerns

In addition to the care burden that comes from being in a close and sometimes exclusive relationship with the PwMS for a long time, the caregivers disclosed their moods and concerns (Fig 2).

Many have reported they have to manage their relatives in complete solitude. If in the initial stage of the disease, in the absence of severe disabilities, some have received closeness and emotional support from family members and friends, as the disease progressed this has occurred less and less. It was especially those who have been in charge of assistance for a long time who reported a condition of neglect by friends and family network. This complaint was

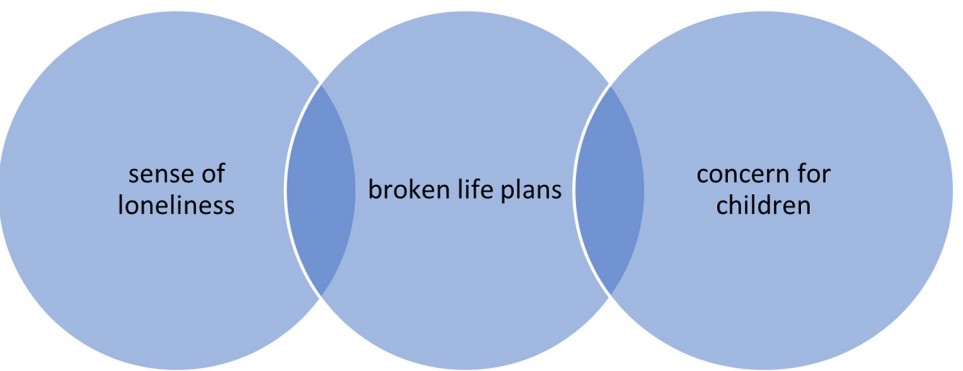

**Fig 2. States of mind and worries.**

often made in resigned and sometimes afflicted tones; at other times, however, tinged by sarcasm and anger.

*"After the diagnosis relatives and friends were supportive, sympathetic. However, lately, we lost our friends, and now we are alone and I feel the complete responsibility . . . I'm alone with my hard caregiving work and I don't have time for myself"* (P10).

In addition, some family members have testified to the suffering inflicted by the illness on their life project, both as regards the relationship with their loved one, and the possibility of dedicating oneself to one's passions, for example, to retirement plans. A wife referred to her prospects of growing old with her husband by cultivating common interests (travelling and hobbies).

*"Since he got MS our life has not been the same. . .When we go on holiday is not easy, he can't walk on the sand, or when we go to the mountains he can't take the chairlift or walk to a hut. . .Our life plans to travel around the world are broken"* (P13).

Another theme that emerged was the concern for children, especially when they are in adolescence or youth. Caregivers recognize how difficult it is for a child to witness the degenerative course of a parent's disease, and how this becomes an element of great relevance both for the emotional-psychological implications, and for those inherent to life planning.

*"I remember when our daughter was 4 years old, and my husband started to have motor impairment, her desire was to run through a park and to ride a bicycle with her dad. . .. and it wasn't easy for her not being able to do everything with him"* (P06). For this reason, many parents have expressed their intention not to lean on their children, preserving them as much as possible from the burden of care. A mother told: *"Our son is present, he is available and he wants to help me. But he has to study and think about his life. . . so I don't want to overload him with the care"* (P14).

## Theme c: Resources to cope with the assistance burden

Many family members have declared that they do not have the resources to counter the daily grind, alluding, as we have seen, to the condition of solitude and the lack of time to devote to themselves.

Others, on the other hand, said they resort to forms of help or the use of expedients / strategies to cope with the burden of care. Specifically, many elements included in the so-called extrinsic resources have emerged (Fig 3).

First of all, family members (children, husbands, sisters, grandchildren), considered an essential presence in the life of the caregiver. Friends and relatives are, in fact, points of

EXTRINSIC RESOURCES
- family members
- friends
- hobbies

INTRINSIC RESOURCES
- love towards the loved one
- empathic understanding
- patience and dedication

**Fig 3. The resources of caregivers.**

reference not only in times of difficulty, as stated by a witness: "*. . .if we have a need, I call my sister, she is the one who supports us in everything (. . ..) she provides both psychological and personal material support, above all, in everyday life chores*" (P06). In fact, the parental or friend network becomes, for the caregiver, a reassuring and helpful presence with which to share the ordinary care burden and find comfort in moments of discouragement. A witness tells us: "*(. . .) we have one granddaughter who makes our days happy, so it's a good thing she's helping us, because I tell her frankly that some mornings I wouldn't even want to get up, but. . .. but in short, we start our day, I think of her and we move forward*" (P13).

Other resources that caregivers cite include: reading, sports and being able to succeed in cultivating their passions; for example: "*At the end of the day, when my husband is in bed, and I'm tired, I immerse myself in reading. . . reading is my greatest resource*" (P15). "*I need to run once a week to recharge my battery*" (P05).

Finally, to cope with the burden of assistance, many family members declare that they refer to intrinsic resources linked: a) to the relationship and love towards one's loved one *"our love is my resource. . ."*(P14), b) to understanding and empathic attitudes towards the disability of one's loved one: "*when I'm tired or angry, I ask myself: and now how does he feel? I think that my ability to put myself in his shoes is my best resource*" (P16). c) to a spirit of dedication and patience. A wife says in this regard: "*Because is there an issue that at least I can elaborate on "it happened to him but what if it had happened to me?" Then you feel energized, start the engine, also because there is little to say, the sorrow is so great, but also the love you feel is so great, and if you didn't love those people, they couldn't handle so many and hard tasks*" (P07).

## Theme d: Take charge by the MS CPW of the AUSL of Bologna

All caregivers are satisfied with the care and assistance offered by operators and by services related to the SM CPW of the AUSL of Bologna. Particular appreciations were made to the Multiple Sclerosis Center, which from the moment of diagnosis takes charge and follows the patient from the neurological, psychological and rehabilitative point of view, remaining a constant point of reference. "*Everything is well organized. The services are linked with a high quality of the care, we feel supported.* (P01).

Particularly kindness, active listening and open dialogue, in addition to the expertise of the healthcare operators are the most appreciated elements. The facility to contact the head of the

Center, by mail or by telephone, at any time of the day to receive clarifications in the case of uncertainties, is an aspect that assumes great value for family members for the sense of security and of taking charge that this entails. "*When we meet the doctor we talk for a long time. She asks me about my husband, his behaviours and she checks the exams with the previous ones. . .the doctor is very helpful and her communication is clear* (P08).

In addition, the possibility of having an integrated pathway between hospital and local services is recognized as one of the strengths of the CPW. Regarding the limits of the pathway, the caregivers almost unanimously lamented the small number of cycles of physiotherapy that the CPW offers. They were being forced to resort to private physiotherapists to respond to the rehabilitation needs of their relatives. To this was added the limitations concerning the waiting list for psychiatric consultations to be carried out locally. Although most of the family members praised the helpfulness of the operators, some of them found the detached attitude of some doctors, a critical element, referring, in particular, to the perceived lack of interest in what happens outside the clinical-therapeutic intervention such as, for example, in the management of home care.

## Theme e: The role of the Patient Association

All caregivers recognized the great value that the local Bologna section of AISM has in supporting them in care burden. The AISM, founded in 1968, works in a structured way to address all issues having an impact on PwMS. Reference was made to the many services that are offered such as: a) the transportation of the patient from home to the hospital in case of medical visits, rehabilitation sessions or to take part to activities at the headquarters of the Association; b) the physiotherapy sessions that take place inside of the headquarters; c) facility that MS patients are given to leave home and get to know other people with the same pathology; d) the opportunity to participate in recreational activities, such as theatre or to receive the support offered by the psychologist present at the section; f) the drug delivery service; g) family counselling and guidance service. In addition to the services offered, family members have expressed great satisfaction for the presence of volunteers, often young, who offer concrete help to support the physical and emotional burden of the caregiver. A family member tells: "*the Association sent young volunteers home to help me, they bring a lot of joy. . . some of them help a lot, for example he moved him* (in reference to the patient), *f, he studied how to do it, in short, they are very good*" (P14). Another caregiver tells with emotion: "*when I see the boys of this association coming, I am really moved, because they are doing everything to help you, and that is exactly what we need, the gesture, and not the thing itself*" (P01). On this line another family member continued: "*that little girl came to me this morning to take me here to the section of the association, and when she entered she said to me: "Good morning V. how are you?", it brightened my day, because hearing someone saying to me "how are you" is really important*" (P10).

## Theme f: The needs of family members

This represents one of the most relevant results of our interviews, as it highlights the mostly unmet needs of caregivers. These needs should be incorporated into the MS CPW, to provide solutions helpful for the quality of life and wellbeing of both the PwMS and their caregivers. The participants' experiences showed that the needs which can relieve the fatigue of assistance are: a) needs regarding the condition of the caregiver, b) needs related to economic and social issues and c) needs related to the assistance of the patient and/or to a better organization of the MS CPW (Fig 4).

As regards the condition of the caregiver, most of the interviewees expressed the need to have time for themselves and someone to share their load with. Some have expressed the need

| needs regarding one's condition as caregiver | social and economic needs | nursing and care needs |
|---|---|---|
| • time for oneself<br>• presence of other significant others to share responsibilities<br>• self-help groups of caregivers to address the issue of disclosing MS diagnosis to children | • economic aid<br>• greater political-administrative commitment | • physiotherapy management |

**Fig 4. Needs expressed, grouped by type of need.**

to be able to consult other caregivers on some problems, such as helping children in adolescence or youth to enter into a relationship with the parent's disease. "*I'd like to have someone helping me for a few hours a day, because I'm constantly restrained at home with my husband. I confess I need to unplug for a bit, and I think that many caregivers have the same need [. . .] I need to go out during the day and take time for myself*" (participant 08); "*For me it is important to share the responsibility of care. . .I'm an only daughter and I need someone to share my doubts, my fears and fatigues*" (P12); Others instead, have expressed the need to receive an economic contribution to support the costs incurred for home assistance by the family to compensate for the reduced earnings deriving from continuous assistance to their family member "*First*, *surely we need a financial support*: *most of the costs of physiotherapy are borne by us*" (participant 11). On the social level, some have expressed the need for greater political-administrative attention paid to the problems faced by disabled people, such as the need to remove architectural barriers, to increase dedicated services and to promote voluntary associations. "[. . .] *we need to sensitize the public administration because there are many diseases like MS*, *we need help from the public administration*, *the Associations for the rights of the patients*, *the rights of caregivers*, [. . .] *laws*, *in particular to remove architectural barriers* (P 02); Finally, most of the family members asked for ongoing physiotherapy sessions for their relative, an issue which had already emerged "*We need to have more physiotherapy sessions*, *especially at home*, *physiotherapy paid for by the Local Health Authority of Bologna*" (P06).

## Discussion

The results that emerged from our study are comparable to those of numerous quantitative [31,32] and qualitative [4] Italian and international studies about the *caregiver burden*. In particular, the results of this study are in line with those of an Italian research conducted through the use of validated scales such as the Health Survey (SF-36) [33], the Caregiver Burden Inventory [34], the Caregiving tasks in multiple sclerosis scale and the Psychological Well-Being Scales [35].

The analysed interviews do not only reveal the paramount importance of the caregiver, but highlight the care burden deriving from the management of the PwMS. Based on one study by Boeije et al. [36], the burden of care is a recognized phenomenon, albeit many times tolerated, as associated with an implicit sense of duty. As stated by the subjects we interviewed, the role of the caregiver is, in fact, complex and articulate. The role does not only involve the fulfilment of physical needs, the need for continuous surveillance in the intermediate and advanced

stages of illness and personal assistance (hygiene and personal care), but also taking charge of the emotional-affective as well as social dimension of one's loved one.

Many interviewees reported that they experience fatigue particularly when it is necessary to carry out hygiene and mobilization activities and it is only partially alleviated by the use of aids, such as the lifter. Caregivers also argue that fatigue increases more rapidly with advancing age: older family members are often no longer able to provide for the physical needs of loved ones and report feeling concerned about their state of health and the consequences it may have on care [22,37].

However, as previously stated, most of the welfare burdens arise from the intangible aspects of care [38]. These include concern about the progress of the disease, the perception of loneliness, alienation, isolation and sense of responsibility for the ordinary management of assistance. There is also a restlessness for the difficult reconciliation between work and assistance, which very often results in the reduction or in the abandonment of work, as also stated by numerous studies [39–43]. Many of the caregivers' stories reveal a sense of melancholy related to the loss of life projects, both for oneself and for the couple. This is combined with concern for the future and a sense of apprehension in the attempt to preserve the children from the assistance burden as much as possible. A deeply felt problem concerns the consequences that a parent's MS has on adolescent children or young adults. The subject is widely addressed in the literature [44]. Due to the emotional burden children suffer the often sudden need to front higher responsibilities and to be sometimes excessively involved in assisting the parent [45,46].

Resources are mobilized to cope with the burden of assistance. Resilience strategies related to two categories: extrinsic resources (family, friends, maintenance of their activities, social welfare activities) and intrinsic resources (love, dedication and empathy), are put in place. Thanks to these, caregivers do not only manage to decrease the burden, but also to make sense out of the process of care in itself.

It is interesting to highlight that social welfare activities are also included among the extrinsic resources offered by the AISM Association. The promotion, guidance and financing of scientific research and the provision of services, strives to improve the quality of life of PwMS by promoting their full inclusion and participation in community life, while also supporting the role of the caregiver. In addition to rehabilitation, drug delivery, transport of the PwMS and psychological and administrative support the caregivers found valuable the ability of young volunteers who enter the home, to create closeness and moral support.

It is important to underline that the AISM section of Bologna participated in the writing and periodic revision of the CPW of the AUSL of Bologna, speaking out for the needs of the PwMS and caregivers present in the territory.

Although limited availability of physiotherapy sessions at home has been reported as a problem at a national level, caregivers, in general, appreciated the services and organization of the MS CPW. Particular value was given to the availability, kindness and ability to take charge of operators. Therefore, the CPW should also be included among the extrinsic resources available to caregivers.

The need to share one's condition as a caregiver, thereby emerging from loneliness and isolation, and the need to devote more time to self-care, as well as the request for more commitment to patient and caregiver welfare policies are in line with what is present in the literature and with the data from the ISTAT-AISM survey of 2017.

Caregivers represent an essential point of reference for the management of PwMS at home. As there is evidence that their quality of life is severely compromised due to a sense of loneliness in the daily care of their loved ones, it is necessary for health professionals, caring for PwMS, to pay particular attention also to the needs of family members, trying as much as possible to offer interventions to support and alleviate their fatigue.

## Limitations

The study had the objective to understand the experience of caregivers of people in the care of the MS CPW of the AUSL of Bologna. One limit of the study is represented by the failure to succeed in recruiting caregivers attending other Patient Associations present in the area or caregivers of people external to the MS CPW to evaluate possible differences between those who were vs were not in charge of the MS CPW.

## Conclusions

As a chronic and often disabling disease, MS does not only have an impact on the individual affected, but also on significant others. The feeling of fatigue, loneliness, alienation and isolation that caregivers experience on a daily basis significantly affects their quality of life and health status. There is a need for caregivers to reconcile working times with care times, to devote more space to self-care and to have moments in which being able to share one's condition. These issues must be at the core of health policies. Solutions are necessary to avoid the occurrence of physical and emotional breakdown which could cause the disruption of the relational balance driving the choice of home care. Consequently it is necessary for health services to take full responsibility not only for the PwMS, but also for their caregivers.

Healthcare professionals should recognize that the care, needs and goals of PwMS and caregivers differ and, therefore, should provide customized educational and supportive strategies, consistent with their specific needs.

The data emerging from this survey also indicates the need to increase the attention paid not only by professionals, but also by institutions, to family members, often forgotten.

Considering that patient associations play a significant role in supporting caregivers, their widespread and active participation in strategic plans is increasingly desirable in the organization of care, as was the case for the MS CPW of the AUSL of Bologna.

The results of this research, especially as regards the issue of family needs, are relevant to pursue an improvement in the care of the family and the PwMS at each stage of the disease.

Further investigations, carried out with a mixed method on the same sample, are warranted to improve the quality of life of PwMS and their caregivers, by devising organizational models and support strategies for healthcare professionals.

## Supporting information

**S1 File.**
(PDF)

## Acknowledgments

The publication of this article was supported by the "Ricerca Corrente" funding from the Italian Ministry of Health.

We wish to thank all the caregivers of PwMS who have participated in this study and the Sezione di Bologna of the Associazione Italiana Sclerosi Multipla for continuous support and collaboration.

## Author Contributions

**Conceptualization:** Stefano Benini, Carlo Descovich.

**Data curation:** Stefano Benini, Erika Pellegrini.

**Formal analysis:** Stefano Benini.

**Investigation:** Erika Pellegrini.

**Methodology:** Stefano Benini.

**Supervision:** Alessandra Lugaresi.

**Writing – original draft:** Stefano Benini.

**Writing – review & editing:** Carlo Descovich, Alessandra Lugaresi.

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
