## [Decision Letter · Decision Letter 0]

26 May 2022

PONE-D-22-05892BURDEN AND RESOURCES IN CAREGIVERS OF PEOPLE WITH MULTIPLE SCLEROSIS: A QUALITATIVE STUDYPLOS ONE

Dear Dr. Lugaresi,

Thank you for submitting your manuscript to PLOS ONE. After careful consideration, we feel that it has merit but does not fully meet PLOS ONE’s publication criteria as it currently stands. Therefore, we invite you to submit a revised version of the manuscript that addresses the points raised during the review process. Both reviewers viewed the merits of your manuscript but they indicated a need for some more details regarding your methodological approach.  As well, Reviewer #2 felt there was a need for some more critical reflection regarding your themes. Please also review the manuscript submission guidelines if you decide to re-submit your paper (e.g., please ensure your submission is double-spaced). 

We look forward to receiving your revised manuscript.

Kind regards,

Sander L. Hitzig

Academic Editor

PLOS ONE

Journal Requirements:

Reviewers' comments:

Reviewer's Responses to Questions

**Comments to the Author**

1. Is the manuscript technically sound, and do the data support the conclusions?

Reviewer #1: Yes

Reviewer #2: Partly

2. Has the statistical analysis been performed appropriately and rigorously? 

Reviewer #1: N/A

Reviewer #2: N/A

3. Have the authors made all data underlying the findings in their manuscript fully available?

Reviewer #1: No

Reviewer #2: Yes

4. Is the manuscript presented in an intelligible fashion and written in standard English?

Reviewer #1: Yes

Reviewer #2: Yes

5. Review Comments to the Author

Reviewer #1: The paper is readable and clear. Thanks for the opportunity of reading about this topic!I want to thank the authors for this exciting work. However, some points should be considered to revise the manuscript.

Methods:

We invite the authors to justify their decision of methodology in further detail. Numerous other methodologies have been used in caregiver research and thus your justification is not suitable. The implications of this approach are warranted.

What does 'All protocol requirements have been formulated with respect for data confidentiality and

34 confidentiality required by the applicable regulatory regulations." mean?

The analysis section needs more detail.

Results

Why are these presented as bullet points? A more throughout thematic analysis of the burden of assistance and the effect of the effort is needed.

Overall:

I think a review for colloquial language should be conducted

Reviewer #2: This is an interesting topic focusing on the burden of caregiving and exploring further what that burden might look like. There is also a brief evaluation of the MS clinical pathway and the local support group.

At the moment, the rationale for the study could be stronger in relation to the gap in the literature and also the positioning of the study in relation to the evaluation context – is this really an evaluation of the MS clinical pathway? If it is then there needs to be a much closer tie between the facets of burden identified and the ways in which the MS clinical pathway does/does not meet those needs currently (maybe a table would help here?)

Introduction

• The rational needs to be clearer. Old studies are cited but what has significantly changed in the past 20 years that suggests that an update is necessary – I agree it might be the context needs to be made explicit.

• If this is going to include an evaluation of the MS clinical pathway then we need to hear more about it – we need some context about what it is/its aims etc

• What is meant in this context by ‘resources’ - groups, family and fiends, online information etc? This should be described further with relevant literature on what we already know about these resources included.

Method

• Justify excluding early stage caregivers

• What does AUSL stand for?

• What is the survey – do you mean the interview schedule?

• It would be useful to have more information around the interview schedule and some example questions? Did these vary for the focus groups?

• How many focus groups were there?

• Describe the characteristics of the caregivers (the participants) first before describing the pwMS and refer to table 1 in the text.

• What does EDSS stand for?

• More information on the analysis is required -how was this conducted, which authors were involved?

Results and discussion

• The themes mentioned in the opening paragraph of the results section do not match the themes presented beneath. Different number (4 then 6) in a different order and with different names. This needs to be reviewed.

• The first theme is interesting and speaks to the complexity of ‘burden’ but it feels a little rushed and under explored here. Can you use longer, richer quotes?

• Quotes need to be attributed to P1, P2 etc.

• Some themes are very descriptive in nature and are list like in places and some contain very few or no quotes to support the interpretation.

• Overall, this section feels unbalanced. The first theme feels like a theme with a lot of potential for really exploring the different facets of burden, the other themes feel either very light weight with few supporting quotes or more like a straightforward evaluation of the service. I think the paper needs to emphasise either the themes of burden further or at least find a way to tie together the burden and the current resources/support of the service to bring the findings together and identify key gaps for service provision and for future research. This could be achieved in the results and also carried through into the discussion.

6. PLOS authors have the option to publish the peer review history of their article (what does this mean?). If published, this will include your full peer review and any attached files.

Reviewer #1: No

Reviewer #2: No

---

## [Author Response · Author response to Decision Letter 0]

5 Jan 2023

RESPONSE TO REVIEWERS

Reviewers' comments:

Reviewer's Responses to Questions

Comments to the Author

Reviewer #1: The paper is readable and clear. Thanks for the opportunity of reading about this topic! I want to thank the authors for this exciting work. However, some points should be considered to revise the manuscript.

Methods:

We invite the authors to justify their decision of methodology in further detail. Numerous other methodologies have been used in caregiver research and thus your justification is not suitable. The implications of this approach are warranted.

We thank the reviewer for the request. We have better explaind our chice in the methods section

What does 'All protocol requirements have been formulated with respect for data confidentiality and

34 confidentiality required by the applicable regulatory regulations." mean?

We thank the reviewer for highlighting the need to clarify this point. We have rewritten the sentence more concisely and clearly. We intended to certify that all regulations on data protection had been thoroughly applied.

The analysis section needs more detail. 

The study is a qualitative work. We have now added more details about the analytical method, adding a sentence at the end of this section: “…The research team used several strategies to ensure the accuracy of the findings. The analytic team met regularly throughout the iterative process of data analysis to reflect on ways individual values, experiences, or preconceptions might influence or create bias in interpretations of the data may have informed perceptions of participant statements and emerging themes. In addition, the analytic team wrote memos throughout the data analysis, noting observations and reactions to the interview responses. Finally, the last author (AL), who had not previously participated in data analysis, conducted an independent review of the findings and coding documents to assess the trustworthiness of the analysis.”

Results

Why are these presented as bullet points? A more throughout thematic analysis of the burden of assistance and the effect of the effort is needed.

Rather than bullet points it was a sequence of the 6 identified relevant themes analyzed from the transcripts of the interviews and focus group. We have maintained, therefore, the division into the main themes emerged from interviews and for each we have tried to extract and cite verbatim meaningful parts of the interviews. We have also tried to deepen the considerations about the meaning of the different issues raised by caregivers as requested.

Overall:

I think a review for colloquial language should be conducted

We thank reviewer #1 for highlighting this point. We have revised language accordingly, and we hope this will satisfy reviewers and editor. However colloquial expressions have been maintained in excerpts from interviews.

Reviewer #2: This is an interesting topic focusing on the burden of caregiving and exploring further what that burden might look like. There is also a brief evaluation of the MS clinical pathway and the local support group.

At the moment, the rationale for the study could be stronger in relation to the gap in the literature and also the positioning of the study in relation to the evaluation context – is this really an evaluation of the MS clinical pathway? If it is then there needs to be a much closer tie between the facets of burden identified and the ways in which the MS clinical pathway does/does not meet those needs currently (maybe a table would help here?)

We thank the reviewer for raising the issue of the evaluation of the clinical pathway. After thorough consideration we have opted for not evaluating the CPW, only mentioning some issues raised by caregivers. We have amended the manuscript eliminating point b from the last paragraph of the introduction.

• The rational needs to be clearer. Old studies are cited but what has significantly changed in the past 20 years that suggests that an update is necessary – I agree it might be the context needs to be made explicit.

We thank the reviewer for this comment. We have updated the literature and hope to have better clarified the rationale.

• If this is going to include an evaluation of the MS clinical pathway then we need to hear more about it – we need some context about what it is/its aims etc

As stated above we have decided to limit the study to cargivers observations. A further study, in the future, might be dedicated to a formal evaluation of changes needed to improve the efficacy of the CPW and improve caregivers satisfaction. A yearly monitoring and revision of the CPW is performed in this respect

• What is meant in this context by ‘resources’ - groups, family and friends, online information etc? This should be described further with relevant literature on what we already know about these resources included.

We have added a sentence in the methods section: b) assess which resources the carers rely on to manage the burden of care. These include: psychological help, emotional resources; relationship (family, friends or groups); support system resources, financial resources etc.

Method

• Justify excluding early stage caregivers

We thank the reviewer for raising this point. In MS caregivers are not needed in the early stages of Multiple Sclerosis, where the patient may have subtle impairment of some functions, but usually does not need a caregiver. Thus we did not exclude caregivers of people in the early stages of the disease, who were very few, if any. We have concentrated our attention on PwMS at stages where impairment and disability are more pronounced and require a caregiver 

• What does AUSL stand for? 

We thank the reviewer for asking. This is the acronym for Azienda Unica Sanitaria Locale, which identifies the Local Health Authority of Bologna, responsible for Healthcare provision in the area of Bologna, in the Emilia Romagna Region, in Italy.

• What is the survey – do you mean the interview schedule?

We thank the reviewer for suggesting a more precise wording.

• It would be useful to have more information around the interview schedule and some example questions? Did these vary for the focus groups?

We thank the reviewer for asking for more information. We have added the required information in the manuscript. In particular the same questions were asked both in interviews and in the single focus group

• How many focus groups were there?

We thank the reviewer for asking. We held a single focus group, due to the small number of subjects included in the study and willing to participate to a focus group. People unable or unwilling to attend the focus group were individually interviewed

• Describe the characteristics of the caregivers (the participants) first before describing the pwMS and refer to table 1 in the text.

We thank the reviewer for noticing that table 1 was unlinked. We have now amended this imprecision and modified the order of presentation of caregiver and patient characteristics according to the reviewer suggestions.

• What does EDSS stand for?

We apologize for the use of a technical acronym without explaining first its meaning

The EDSS is the Expanded Disability Status Score, a scale used to assess impairment (below 4) or disability (4 or higher). The score derives from the scores in 7 different “functional systems”: pyramidal, sensory, brainstem, cerebellar, visual, mental, bowel and bladder. It’s not arithmetical, 0 stands for normal neurological examination, 10 for death due to MS. In between relevant scores for this study: 6 stands for needing a stick, 7 needing a wheelchair (Kurtzke, 1983:

Kurtzke JF. Rating neurologic impairment in multiple sclerosis: an expanded disability status scale (EDSS). Neurology. 1983 Nov;33(11):1444-52. doi: 10.1212/wnl.33.11.1444. PMID: 6685237).

• More information on the analysis is required -how was this conducted, which authors were involved?

Authors involved in interviews and the focus group were SB and EP. CD provided critical methodological supervision and AL discussed the results with the other co-authors and revised the draft. Details have now been reported in the last paragraph of the analysis section.

Results and discussion

• The themes mentioned in the opening paragraph of the results section do not match the themes presented beneath. Different number (4 then 6) in a different order and with different names. This needs to be reviewed.

We thank the reviewer for noticing the discrepancies between the methods section and the results section. We have revised the text accordingly

• The first theme is interesting and speaks to the complexity of ‘burden’ but it feels a little rushed and under explored here. Can you use longer, richer quotes?

Thank you for asking longer quotes, we had omitted for brevity. We have now widened the quotations

• Quotes need to be attributed to P1, P2 etc.

We have now attributed quotes to individual caregivers, in the table called PARTICIPANTS, therefore in brief P (P01 to P17)

• Some themes are very descriptive in nature and are list like in places and some contain very few or no quotes to support the interpretation.

We have increased quotations in order to provide a better support for themes identified

• Overall, this section feels unbalanced. The first theme feels like a theme with a lot of potential for really exploring the different facets of burden, the other themes feel either very light weight with few supporting quotes or more like a straightforward evaluation of the service. I think the paper needs to emphasise either the themes of burden further or at least find a way to tie together the burden and the current resources/support of the service to bring the findings together and identify key gaps for service provision and for future research. This could be achieved in the results and also carried through into the discussion.

This is a very relevant observation. We hope the revised version of the results and discussion sections will be found satisfactory

6. PLOS authors have the option to publish the peer review history of their article (what does this mean?). If published, this will include your full peer review and any attached files.

Do you want your identity to be public for this peer review? For information about this choice, including consent withdrawal, please see our Privacy Policy.

Reviewer #1: No

Reviewer #2: No

---

## [Decision Letter · Decision Letter 1]

6 Mar 2023

PONE-D-22-05892R1BURDEN AND RESOURCES IN CAREGIVERS OF PEOPLE WITH MULTIPLE SCLEROSIS: A QUALITATIVE STUDYPLOS ONE

Dear Dr. Alessandra Lugaresi,

Thank you for submitting your manuscript to PLOS ONE. After careful consideration, we feel that it has merit but does not fully meet PLOS ONE’s publication criteria as it currently stands. Therefore, we invite you to submit a revised version of the manuscript that addresses the points raised during the review process.

1. Clearly provide details of the Ethical Review Board the protocol was submitted to

2. Check journal requirements for labelling tables

3. Remove bullet points in discussing 'Theme a'

4. Change the word to 'Limitations' on p13

5. Define the appropriate referencing style in line with journal requirements and ensure consistency. Each reference is presented differently from the next.==============================

We look forward to receiving your revised manuscript.

Kind regards,

Habil Otanga, Ph.D

Academic Editor

PLOS ONE

Journal Requirements:

Additional Editor Comments (if provided):

1. Clearly provide details of the Ethical Review Board the protocol was submitted to

2. Check journal requirements for labelling tables

3. Remove bullet points in discussing 'Theme a'

4. Change the word to 'Limitations' on p13

5. Define the appropriate referencing style in line with journal requirements and ensure consistency. Each reference is presented differently from the next.

Reviewers' comments:

Reviewer's Responses to Questions

**Comments to the Author**

1. If the authors have adequately addressed your comments raised in a previous round of review and you feel that this manuscript is now acceptable for publication, you may indicate that here to bypass the “Comments to the Author” section, enter your conflict of interest statement in the “Confidential to Editor” section, and submit your "Accept" recommendation.

Reviewer #2: All comments have been addressed

Reviewer #3: (No Response)

2. Is the manuscript technically sound, and do the data support the conclusions?

Reviewer #2: Yes

Reviewer #3: Yes

3. Has the statistical analysis been performed appropriately and rigorously? 

Reviewer #2: N/A

Reviewer #3: Yes

4. Have the authors made all data underlying the findings in their manuscript fully available?

Reviewer #2: Yes

Reviewer #3: Yes

5. Is the manuscript presented in an intelligible fashion and written in standard English?

Reviewer #2: Yes

Reviewer #3: Yes

6. Review Comments to the Author

Reviewer #2: (No Response)

Reviewer #3: 1. Ethical considerations: Clearly provide details of where the protocol was submitted for review (and found that no authorization was necessary)

2. Check journal rules concerning labelling tables

3. Results: Present findings of 'Theme a' without bullet points.

4. Change the word to 'Limitations' on p13

5. References: Define the referencing style and ensure consistency. Each reference is presented differently from the next.

7. PLOS authors have the option to publish the peer review history of their article (what does this mean?). If published, this will include your full peer review and any attached files.

Reviewer #2: No

Reviewer #3: No

---

## [Author Response · Author response to Decision Letter 1]

26 Mar 2023

PONE-D-22-05892R1

BURDEN AND RESOURCES IN CAREGIVERS OF PEOPLE WITH MULTIPLE SCLEROSIS: A QUALITATIVE STUDY

PLOS ONE

Point-by-point response

1. Clearly provide details of the Ethical Review Board the protocol was submitted to

We have added in the text what was already attached to the submission: the local ethics committee is the 

Comitato Etico di Area Vasta Emilia Centro della Regione Emilia-Romagna (CE-AVEC), and was active since 

10/1/2018. As stated in Italian, there is no need to submit to the ethics committee, as no personal details have been provided

2. Check journal requirements for labelling tables

We have adhered to journal requirements (Caption above the table). We apologize for not noticing earlier.

3. Remove bullet points in discussing 'Theme a'

We have removed the bullet points, numbering in the text the 6 issues

4. Change the word to 'Limitations' on p13

We have changed limits to LIMITATIONS, as required

5. Define the appropriate referencing style in line with journal requirements and ensure consistency. Each reference is presented differently from the next.

We apologize for the imprecisions. References have been revised to be in line with journal requirements

In addition we have corrected some typos we picked in re-reading the manuscript.

Journal Requirements:

We have not cited papers which have been retracted. A few are not on PubMed, but have a DOI, which we have used to enable readers to easily find the right article or chapter in a book (they are open access).

Additional Editor Comments (if provided):

1. Clearly provide details of the Ethical Review Board the protocol was submitted to

2. Check journal requirements for labelling tables

3. Remove bullet points in discussing 'Theme a'

4. Change the word to 'Limitations' on p13

5. Define the appropriate referencing style in line with journal requirements and ensure consistency. Each reference is presented differently from the next.

See above

---

## [Editor Report · Decision Letter 2]

27 Mar 2023

BURDEN AND RESOURCES IN CAREGIVERS OF PEOPLE WITH MULTIPLE SCLEROSIS: A QUALITATIVE STUDY

PONE-D-22-05892R2

Dear Dr. Alessandra Lugaresi,

We’re pleased to inform you that your manuscript has been judged scientifically suitable for publication and will be formally accepted for publication once it meets all outstanding technical requirements.

Kind regards,

Habil Otanga, Ph.D

Academic Editor

PLOS ONE
---

## [Editor Report · Acceptance letter]

6 Apr 2023

PONE-D-22-05892R2 

Burden and resources in caregivers of people with multiple sclerosis: a qualitative study 

Dear Dr. Lugaresi:

I'm pleased to inform you that your manuscript has been deemed suitable for publication in PLOS ONE. Congratulations! Your manuscript is now with our production department. 

Kind regards, 

on behalf of

Dr. Habil Otanga 

Academic Editor

PLOS ONE